# Sustainable and Nutritional Recommendations for the Development of Menus by School Food Services in Spain

**DOI:** 10.3390/foods11244081

**Published:** 2022-12-16

**Authors:** María Asunción Martínez-Milán, María Carmen Davó-Blanes, Iris Comino, Pablo Caballero, Panmela Soares

**Affiliations:** 1Department of Community Nursing, Preventive Medicine and Public Health and History of Science, University of Alicante, 99-03080 Alicante, Spainpanmela.soares@ua (P.S.); 2Public Health Research Group, University of Alicante, 99-03080 Alicante, Spain

**Keywords:** school feeding, health promotion, school food services, nutritional guidelines, food sustainability

## Abstract

Background: Recommendations for the development of school menus can promote healthier, more sustainable food systems. In Spain, these recommendations depend on regional governments (Autonomous Communities—CCAAs) that develop their own guides. The objective of this study was to explore the nutritional and sustainability recommendations for the development of menus by school food services in Spain. Methods: Guides were reviewed that were available on the official web pages of the councils of health and education. Twenty-four variables were studied and organized into three categories: characteristics, nutritional recommendations and sustainability. The number of recommendations included in each guide was counted. The weekly frequency of the suggested food provision for each food group was calculated, as was the average, median, standard deviation, confidence interval and interquartile index. Results: Overall, 13 guides were reviewed from different CCAAs. All of them included at least three of the nutritional recommendations, two suggested restrictions in the provision of foods with high quantities of salt and six suggested restrictions in foods with high levels of trans and saturated fats and sugars. All except one guide recommended the weekly provision of foods by food group: protein-rich foods (n = 8), cereals and root vegetables (n = 6), vegetables (n = 5.2) and fruit (n = 4.3). Of the eight criteria for sustainability studied, nine guides included one or none. Conclusions: Guides for the provision of meals at school in Spain promote the incorporation of healthy foods; however, they rarely restrict foods with high levels of fat, salt and sugar, and the promotion of food sustainability is only just beginning. These guides should be reviewed and updated to include recommendations that promote healthy and sustainable food systems.

## 1. Introduction

The prevalence of childhood obesity and overweight in Spain is among the highest in Europe, at 18.3 percent and 10.3 percent, respectively [1]. Obesity and overweight are risk factors for the development of chronic and non-transmissible diseases that are among the primary causes of death in the world [2,3]. The increase in overweight and obesity is related to changes in dietary patterns, in which there is a greater intake of processed foods and foods of low nutritional quality and less consumption of fruit, vegetables and legumes [4].

Promoting healthy food among the child population is important, given that dietary habits acquired in childhood tend to be maintained until adulthood [5]. The European Union’s Action Plan on Childhood Obesity 2014–2020 urges countries to address childhood obesity in their policies [6]. Schools are considered key environments in achieving this objective [7,8], and therefore, one of the aims of the World Strategy on Diet, Physical Activity and Health is to promote the provision of healthy foods in schools [9].

School food services provide a space in which to promote healthy nutritional habits [10,11,12]. Different countries have developed food programs in schools, in which approximately 638 million children around the world receive meals [13]. In Spain, the number of schools that offer meal services was approximately 17,221 during the 2019–2020 school year, with around 2,256,894 children served each day [14]. It is estimated that approximately 40 percent of enrolled students in preschools and primary schools use cafeteria services in Spain. This value varies from 20 percent to 70 percent by the region of the country [15].

School food plays an important role in the nutrition and health of the child population. The World Health Organization (WHO), together with the United Nations Agricultural Organization, recommends the development of guidelines to support healthy meals in educational centers [8,16]. These guidelines establish the quantity and minimum quality of school meals. These guidelines could be an effective tool to improve the system for food provision in schools. On the other hand, the inclusion of sustainability criteria in food meal programs through governmental plans and programs can support achieving the sustainability objectives established in the 2030 Agenda [17,18,19,20].

Countries including Portugal, the United Kingdom and France have established mandatory nutritional guidelines for the preparation of school meals, while in Italy, Germany and Spain, these guidelines are voluntary, and sometimes there is even variation within countries [8,21,22,23,24]. The implementation of these recommendations in schools could have a positive impact on the availability and consumption of food in the school population [25,26].

In 2010 the Consensus Document on Meals in Educational Centers was produced in Spain (DCSECE) [27], which, following international guidelines, included recommendations on the provision of food groups that promote health and nutrition in school menus. However, the management of school food services in Spain is decentralized, which generates a diverse context given that each Autonomous Community (CCAA) can create its own guides with recommendations for the development of school menus. Considering the potential of the guides to promote the construction of healthy and sustainable food systems, the objective of this study was to explore the sustainable and nutritional recommendations for the development of school menus by school food services in the different regions of Spain.

## 2. Materials and Methods

This was a descriptive transversal, quantitative study carried out through an analysis of guidelines for school food services in Spain’s different CCAAs.

Guidelines were obtained through the official webpages of the Health and Education Councils of each CCAA in June 2019. The analysis included guidelines with recommendations for the development and/or evaluation of menus in school food services. When two sets of guidelines were found for a single CCAA, the most recent one was included, or that which presented recommendations on food provision on menus.

In order to extract the data, an ad-hoc protocol was developed based on the research team’s experience with prior projects related to school feeding and sustainability [12,28]. The WHO nutritional recommendations were used as a reference point [8], as were the guidelines of the Consensus Document on Meals in Educational Centers [27] and the food sustainability criteria of the FAO [16,17]. The protocol collected information on 24 dichotomous variables (yes/no) divided as follows: 1. characterization of the guidelines (n = 3; CCAA, publication date and organization); 2. nutritional recommendations (n = 13); and 3. sustainability criteria (n = 8). The variables related to nutritional recommendations were divided into 3 groups: energetic distribution (n = 4), provision of foods (n = 9) and other recommendations (n = 3; (Table 1 and Table 4). Furthermore, to explore the provision of foods suggested for menu development, we extracted the weekly frequency established for each food group, using as a reference the mid-point in the range of values along which each guide’s weekly proposed menu offering was ranked (Table 2 and Table 3). The analysis of the weekly provision suggested for each food group was carried out considering the proposals of prior studies [28]. Thus, the foods were ranked according to their nutritional values, as follows: (a) recommended foods (healthy): vegetables; cereals and root vegetables (rice, pasta, potato, bread); protein-rich foods (legumes, meat, fish, eggs and dairy products); fruit and nuts; and (b) controlled foods (foods with high levels of salt, sugar and saturated and trans fats): foods with high sugar content (sugary desserts, conserved fruits and juice) and foods with high saturated and trans fat content (fried foods, fatty sauces, pizza, processed and processed meats).

Data were registered in electronic spreadsheets and exported to the statistical programs SPSS and R. A descriptive analysis was carried out. The number of recommendations (nutritional and those related to food sustainability criteria) included in each of the studied guidelines was counted. Furthermore, to study the weekly frequency established for each food group, the average, median, standard deviation, 95% confidence intervals and interquartile index using box plots were calculated.

## 3. Results

Sixteen guides were located with recommendations for school food services. After applying the inclusion criteria, 13 guides were reviewed from 17 CCAAs: Galicia [29], Asturias [30], Basque Country [31], Catalonia [32], Valencian Community [33], Murcia [34], Madrid [35], Castilla and Leon [36], Castilla la Mancha [37], Extremadura [38], Aragon [39], Andalusia [40] and Navarra [41]. The majority were developed by the councils of health (n = 9) and education (n = 4), and more than half were published after the year 2010.

Table 1 shows the nutritional recommendations included in the guides for school food services in Spain by CCAA. Of the 13 studied guides, all included at least three of the nutritional recommendations, and eight guides included half or more than half of them. It can be observed that, by category, the recommendations on energy distribution, raw and net weight for servings by food group and by age group, and the energetic proportion that should be included in the mid-day meal were the most frequent in the guides (n = 10 and n = 9, respectively).

In terms of the recommendations on the provision of foods, seven of the guides included more than half or half of those included in this category; the use of olive oil for food preparation or garnish was the most frequent (n = 13). However, the restriction of foods with high levels of salt was recommended in two of the 13 guides, and both the restriction of foods with high levels of saturated and trans fats and that related to sugar products were included in six of them. All of the guides, except for one, included recommendations for the weekly provision of food by food groups.

Also, all of the guides included at least one of the three other additional recommendations included in this category. Worth highlighting is the provision of information on the school menu to families (n = 10).

Table 2 shows the weekly frequency of the provision of foods (healthy) suggested in the food guides for school food services in Spain. It can be observed that the majority of the guides suggest providing all of the recommended food groups, with great variability, except for the case of nuts, which were suggested in only two of the guides.

**Table 1 foods-11-04081-t001:** Nutritional recommendations included in food guides for school food services by Autonomous Community (CCAA) in Spain.

CCAA = 13														
	GL	PA	PV	CA	CV	MR	MD	CL	CM	EX	AR	AN	NV	Total
NUTRITIONAL RECOMMENDATIONS = 13	5	9	4	3	11	4	9	8	7	5	8	11	9	
Energy Distribution = 4	1	1	2	0	4	1	3	3	4	2	3	4	1	
Proportion of energy that should be included in the mid-day meal for different school ages			X		X		X	X	X	X	X	X	X	9
Distribution in % of macronutrients (carbohydrates, protein and fats)					X			X	X	X		X		5
Energy quantity in kilocalories in food by age group					X		X		X		X	X		5
Weight in raw and net for the rations by food groups and age groups	X	X	X		X	X	X	X	X		X	X		10
**Provision of foods = 9**	**4**	**8**	**2**	**3**	**7**	**3**	**6**	**5**	**3**	**3**	**5**	**7**	**8**	
Does not include root vegetables in the group of vegetables	X	X			X		X	X	X			X	X	8
Provision of vegetables in the form of salad					X	X	X				X	X		5
Includes a specific recommendation for different types of meat (beef, pork, poultry, duck)		X					X				X		X	4
Olive oil for food preparation or garnish	X	X	X	X	X	X	X	X	X	X	X	X	X	13
Restriction of foods with high quantities of saturated or trans fat		X			X			X		X		X	X	6
Restriction of foods with high quantities of salt		X											X	2
Restriction of fried or breaded foods	X	X		X	X		X				X	X	X	8
Restriction of sugary products		X			X			X		X		X	X	6
Inclusion of recommendations for the weekly provision of foods	X	X	X	X	X	X	X	X	X		X	X	X	12
**OTHER RECOMMENDATIONS = 3**	**1**	**1**	**1**	**2**	**3**	**2**	**2**	**1**	**3**	**1**	**2**	**1**	**2**	
Single course option				X	X				X		X		X	5
Menus adapted to people with special food needs			X		X	X	X		X		X		X	7
Information on the school menu for families	X	X		X	X	X	X	X	X	X		X		10

Legend: GL = Galicia; PA = Principiate of Asturias; PV = Basque Country; CA = Catalonia; CV = Valencian Community; MR = Murcia; MD = Madrid; CL = Castilla and Leon; CM = Castilla la Mancha; EX = Extremadura; AR = Aragon; AN = Andalusia; NV = Navarra.

**Table 2 foods-11-04081-t002:** Weekly frequency for the provision of food groups (healthy) suggested in the guides for school food services in Spain.

Recommended Foods	Guides (n)	$\bar{X},\left(Sd\right)$	95%CI	Md	(IIQ)	Boxplot
Vegetables	12	5.2 (2.5)	(3.8;6.6)	5.2	3.0	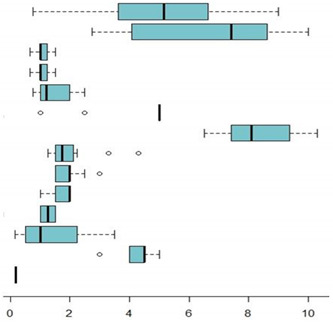
Cereals and root vegetables	12	6.7 (2.5)	(5.3;8.1)	7.4	4.6	
Rice	11	1.1 (0.2)	(0.9;1.2)	1.0	0.2	
Pasta	11	1.1 (0.2)	(0.9;1.2)	1.0	0.2	
Potatoes	12	1.5 (0.6)	(1.1;1.8)	1.2	1.0	
Bread	9	4.3 (1.5)	(3.3;5.3)	5.0	0.0	
Protein-rich foods	12	8.3 (1.3)	(7.6;9.0)	8.1	2.0	
Legumes	12	2.1 (0.9)	(1.5;2.6)	1.8	0.6	
Meat	12	2.0 (0.5)	(1.7;2.2)	2.0	0.5	
Fish	12	1.7 (0.4)	(1.5;2.0)	2.0	0.5	
Eggs	11	1.2 (0.2)	(1.1;1.4)	1.3	0.5	
Dairy Products	12	1.4 (1.0)	(0.8;2.0)	1.0	1.75	
Fruit	12	4.3 (0.5)	(4.0;4.5)	4.5	0.5	
Nuts	2	0.2		0.2	0.0	

The average recommended proposal for protein-rich foods was more than eight times per week, ranging between 6.75 and 10.3. Once the analysis was stratified by the type of protein-rich foods, the average provision suggested for legumes, meat and fish was two times per week, and for eggs and dairy products, it was more than one per week. Of these foods, greater heterogeneity was observed in the weekly provision recommended for legumes, meat, fish and dairy products, while the provision of eggs was homogeneous in all of the guides studied.

In terms of vegetables, the average provision suggested was 5.2 times per week (between 0.75 and 9), while the average for cereals and root vegetables was higher six 6 times per week. Of the foods included in this group, almost all of the guides suggested the same weekly provision of bread (4.3 times per week). For the rest of the foods (rice, pasta and potatoes), the average recommended provision was once per week, with variation among the guides.

In the case of fruit, the weekly provision ranged from three to five times per week, and for nuts, it was lower, at less than once per week.

Figure 1 shows the weekly frequency of the provision for the group of recommended foods proposed by each CCAA in the food guides for the school food services. It shows great variability in terms of the frequency of the weekly provision proposed for the different recommended foods in the different guides.

Table 3 shows the weekly frequency of the provision of controlled food groups suggested in the guides for menu development by school food services in Spain. It can be observed that more than half of the guides do not restrict the provision of products with high saturated/trans-fat content, and less than half restrict products with high sugar content (nine and three guides, respectively).

The average provision for the group of foods with high saturated and trans-fat content is over three times per week. After segregating this food group, it can be observed that the average provision proposed for fruit was around two times per week, slightly higher than the once per week for pre-cooked foods, once per week for fatty sauces, and less than once per week for pizza and processed meats. The boxplot shows that there is more heterogeneity for fried foods in the average weekly provision, as suggested by the studied guides.

In terms of the group of food with high sugar content, the average provision proposed is around once per week. After segregating this food group, it can be observed that for sugary desserts, conserved fruit and juices, the average recommended weekly provision is less than once per week. The boxplot shows little variability in the guides regarding the average weekly provision suggested for this food group.

Table 4 shows the food sustainability criteria included in the food guides for school food services by CCAA. Of the eight criteria included, four guides include five or six criteria, and the rest include one or none. The inclusion of fresh and seasonal products stands out (n = 9), and the seasonality of foods (n = 5) stands out. The decrease in the provision of meat was included in one guide, and the criterion related to fair trade was not included in any guide.

**Table 3 foods-11-04081-t003:** Weekly frequency in the provision of controlled food groups as suggested by Autonomous Communities (CCAAs) for menu development by school food services in Spain.

Controlled Foods	Guides (n)	$\bar{X},\left(Sd\right)$	95%CI	Md	(IIQ)	Boxplot
High sugar content	3	1.1	[0.5–1.6]	1.2	0.5	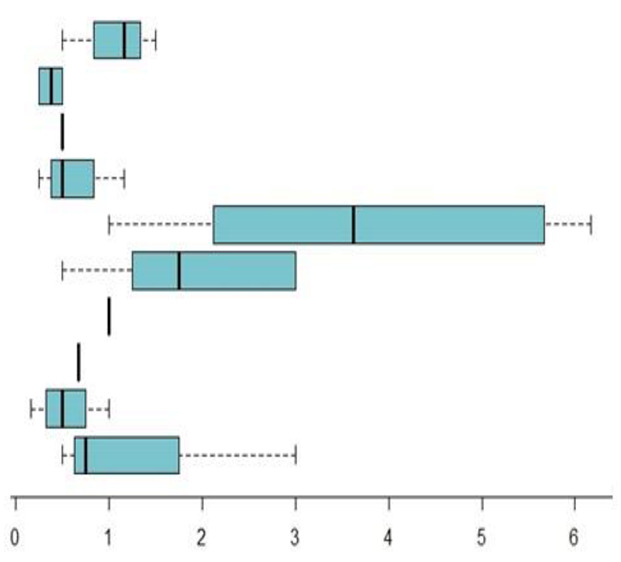
Sugary desserts	2	0.4	[0.2–0.6]	0.4	0.25	
Conserved fruit	1	0.5	-	0.5	0.0	
Juices	3	0.6	[0.2–0.8]	0.5	0.5	
High saturated/trans fat content	8	3.3	[1.1–4.4]	3.6	3.5	
Fried foods	8	1.9	[1–2.9]	1.8	1.8	
Fatty sauces	1	1.0	-	1	0.0	
Pizza	1	0.7	-	0.7	0.0	
Processed meats	7	0.5	[0.2–0.7]	0.5	0.4	
Pre-cooked	7	1.3	[0.3–1.6]	0.8	1.1	

**Table 4 foods-11-04081-t004:** Food sustainability criteria included in the food guides for school food services by Autonomous Community (CCAA).

CCAA = 13														
	GL	PA	PV	CA	CV	MR	MD	CL	CM	EX	AR	AN	NV	TOTAL
FOOD SUSTAINABILITY CRITERIA = 8	2	1	0	5	6	1	1	1	1	1	1	0	6	
Seasonality of foods	X			X				X	X				X	5
Fresh and seasonal products	X	X		X	X	X	X			X	X		X	9
Organic foods				X	X								X	3
Locally sourced foods				X	X								X	3
Locally purchased foods				X	X								X	3
Fairtrade foods														0
Reference to food sustainability in the guide					X								X	2
Decrease in the provision of meat					X									1

Legend: GL = Galicia; PA = Principiate of Asturias; PV = Basque Country; CA = Catalonia; CV = Valencian Community; MR = Murcia; MD = Madrid; CL = Castilla and Leon; CM = Castilla la Mancha; EX = Extremadura; AR = Aragon; AN = Andalusia; NV = Navarra.

## 4. Discussion

This study explored the sustainability and nutritional recommendations for the development of menus by school food services in Spain. Although the guides do not follow a homogeneous pattern, the majority do not include recommendations for restricting foods with high salt content, saturated/trans-fat content or sugars and suggest the provision of protein-rich foods, cereals and root vegetables over fruit and vegetables. It should be noted that the incorporation of sustainability criteria in food guides is only beginning to emerge.

Our results show that the guides do not maintain the same criteria in terms of nutritional recommendations for the development of school menus. In general, they provide information about food groups and age, as well as raw and net weight for portion sizes and the proportion of energy that the mid-day meal should contain. However, not all the guides include recommendations on energetic distribution. The fact that there are no standards related to this area in terms of school menus could result in too little or too many calories provided in the mid-day meal. It is generally considered that the mid-day meal should represent around 35 percent of daily energetic needs [27].

In contrast to the nutritional recommendations for school foods in countries such as the UK, Italy, France, Finland, Slovenia, USA and Brazil [19,23,42,43,44,45,46,47], in Spain, there are few restrictions in the guides related to the provision of foods with high quantities of salt, saturated and trans fats and sugar. This could be due to the fact that guidelines established at the national level refrain from restricting these foods [27]. The consumption of foods with high energy density and sugar is associated with non-transmissible chronic diseases [47]. Taking into account the fact that the habits that are developed in childhood tend to persist into adulthood and can have health consequences [3], one of the recommendations of the WHO is the reduction in the provision of these foods in school food services [8]. Given that the objective of the guides is the promotion of a healthy diet [48], their recommendations should limit the consumption of foods with high quantities of saturated and trans fats, sugar and salt.

The studied guides propose the provision of protein-based foods that is above that of fruit and vegetables. The consumption of fruit and vegetables in the child and adolescent population in Spain is among the lowest in Europe [49], while the consumption of animal protein among the Spanish school-aged population is excessive [48,50]. Investing in the provision of these foods in schools could contribute to a healthier, more sustainable diet. There is evidence of the benefits of the consumption of vegetable-origin foods, as well as fruit and vegetables, for health [51]. These foods are good sources of vitamins and minerals, and consuming them prevents micronutrient deficiency. It also contributes to maintaining healthy body weight and protects against the development of chronic diseases [49]. On the other hand, consuming animal protein is linked to the development of chronic disease [52] and a greater environmental impact [17,53]. Given that providing food on school menus is a strategy to promote changes in eating patterns in the child population [49,54,55]. Including recommendations to increase the provision of fruit and vegetables and decreasing the provision of animal-origin proteins could support diet practices that are healthier and more respectful of the environment [53].

Despite the efforts of international organizations to promote more sustainable food systems through school menus [16,56], the incorporation of sustainability criteria has only recently begun. As shown in our results, it seems that the inclusion of fresh and seasonal products in school menus is being promoted, while the same is not the case for locally-produced, organic or fair trade products. This could be due to the fact that the debate around including sustainability criteria in school food guides is relatively recent. In fact, a study carried out in 2016 showed that few countries incorporate sustainability recommendations in their school food guides [57]. Evidence suggests that incorporating these criteria benefits not only more healthy food provision in schools, it also supports the environment, the economy and the agriculture of the region [17,19]. For this reason, countries such as Italy and Brazil promote sustainability criteria in school food programs by promoting purchasing organic products from local producers [46,58]. In the same way, the World Food Program supports the inclusion of locally-grown products in school food programs in different countries. These actions can contribute to attaining Sustainable Development Objectives (ODS) [59]. The inclusion of sustainability criteria in guides for school food programs is necessary to build food systems that are more respectful of the environment.

Limitations: In interpreting our results, it should be taken into consideration that they are based exclusively on secondary data available on the official CCAA web pages. No other information sources for the different CCAAs were used. However, this information represents a point of departure to identify the sustainability and nutritional recommendations suggested by guides for school food services in Spain to implement healthy and sustainable food programs in schools.

## 5. Conclusions

The guides for the development of menus by school food services in Spain promote the incorporation of healthy foods. However, they do not establish recommendations to restrict foods with high levels of fats, salt and sugar, and recommendations regarding food sustainability are recent. Given that school food guides can promote healthier and more sustainable food systems, in Spain, such guides should be reviewed and updated to:-Increase the provision of fruit and vegetables;-Restrict the provision of products with high quantities of fats, salt and sugars;-Reduce the provision of animal-origin proteins;-Incorporate the direct purchase of foods from local producers and those that come from more sustainable production systems.

## Figures and Tables

**Figure 1 foods-11-04081-f001:**
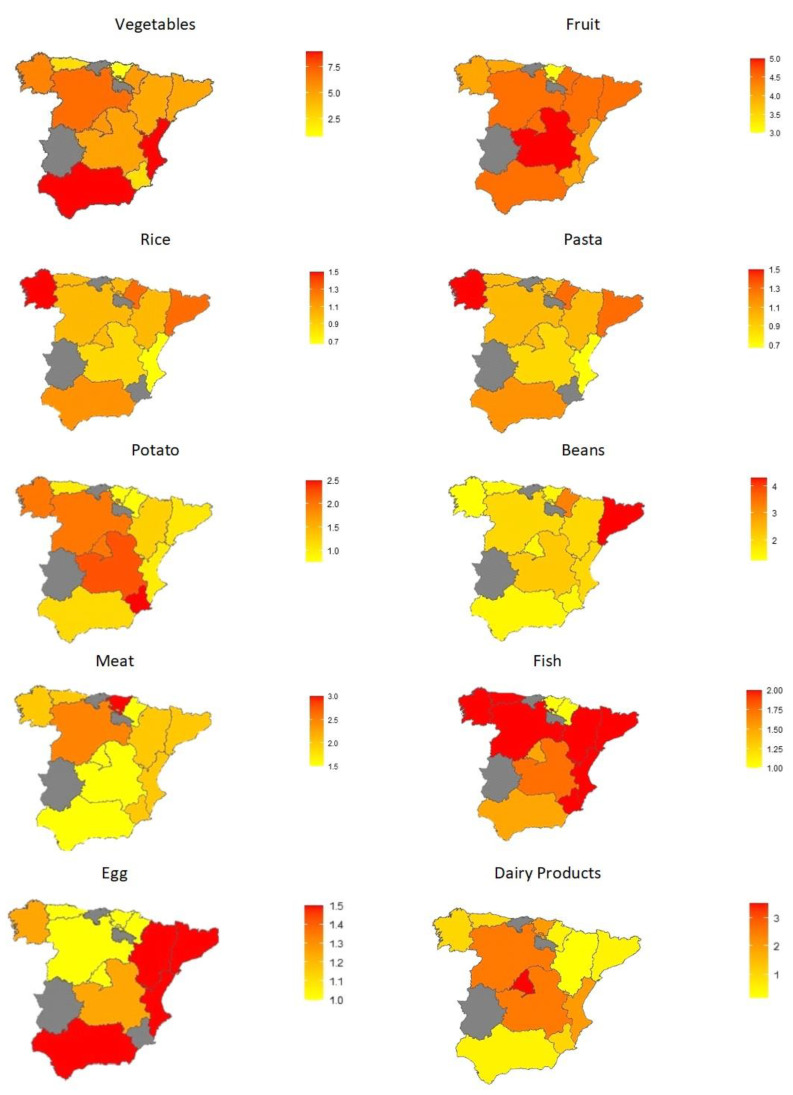
Frequency of the weekly provision of recommended food groups as suggested by Autonomous Communities (CCAAs) in food guides for school food services in Spain.

## Data Availability

The data that support our results can be found on the official webpages of the councils of education and health of the Autonomous Communities of Spain.

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
