# Peer review of "Sustainable and Nutritional Recommendations for the Development of Menus by School Food Services in Spain"

_foods, 2022, doi:10.3390/foods11244081_

Round 1

Reviewer 1 Report

Authors have made good effort in writing this manuscript.

Please rephrase your title little bit if possible “Nutritional and Sustainability” should be replaced with Nutritional and Sustainable.

Please add some more data in background of abstract and that should reflect some background information.

In objective you have just mentioned your topic please add proper objectives in sentence form.

Authors have written very small sentence in abstract section like “A descriptive analysis 17
was carried out., 13 guides were studied., please write in technical ways

Add your recommendations in conclusion of abstract.

Please give rationale of the study in the end of introduction along with proper reasoning of this study.

There are a lot of grammatical mistakes please carefully read the whole article to improve the quality of article.

Statistical analysis is not clear, please mention which design has been used.

Author have used subjective nouns in the study that should be replaced by common nouns.

Author should give collective graphical representation of the results.

Discussion have not been properly compared and no justification have been provided in this section.

Please recheck references according to journal format.

Author Response

Reviewer #1

1. Authors have made good effort in writing this manuscript.

Response: Thank you for this comment and the effort involved in reviewing the manuscript.

2. Please rephrase your title little bit if possible “Nutritional and Sustainability” should be replaced with Nutritional and Sustainable.

Response: Thank you for the observation. In the new version we have adapted the title.  See page 1 line 2

3. Please add some more data in background of abstract and that should reflect some background information.

Response: Taking into account the comment from the reviewer, we have included more information in the summary.  See page 1, summary paragraph 1

4. In objective you have just mentioned your topic please add proper objectives in sentence form.

Response: We have changed the writing of the objective in accordance with the suggestion. See page 1, summary paragraph 1 line 14-15

5. Authors have written very small sentence in abstract section like “A descriptive analysis 17 was carried out. 13 guides were studied., please write in technical ways”

Response: In accordance with the suggestion, we have modified the summary and added more information on the methodology. See page 1 summary paragraph 1 line 16-21

6. Add your recommendations in conclusion of abstract.

Response: In accordance with the suggestion, we have included recommendations at the end of the summary.  See page 1, summary paragraph 1 line 29-31.

7. Please give rationale of the study in the end of introduction along with proper reasoning of this study.

Response: In the new version of the manuscript, we have justified the rationale of the study, at the end of the introduction. See page 3 paragraph 1

8. There are a lot of grammatical mistakes please carefully read the whole article to improve the quality of article.

Response: Thank you for this observation. The manuscript has been reviewed by a native translator.

9. Statistical analysis is not clear, please mention which design has been used.

Response: This was a transversal study, and a descriptive analysis was carried out. We have included information in this new version. See page 3 paragraph 2 line 85  

10. Author have used subjective nouns in the study that should be replaced by common nouns.

Response: In response to the comment from the reviewer, we have modified the writing of the manuscript.

11. Author should give collective graphical representation of the results.

Response: The results of our study are represented graphically in tables 1, 2 and 3. Also, tables 1 and 3 show a boxplot to represent the frequency of the weekly provision by food group for both recommended foods and controlled foods. There is also a graphical representation in terms of maps of the weekly frequency for the different regions of Spain in Figure 1. See pages 5,6,8,9,10,12    

12. Discussion have not been properly compared and no justification have been provided in this section.

Response: In accordance with the suggestion of the reviewer, in the discussion we have incorporated more information that compares our results with those of other studies. See page 14 paragraph 4 line 272-272 and page 15 paragraph 2 line 308-310  

13. Please recheck references according to journal format.

Response: The bibliography has been revised and adapted to the norms of the journal.

Reviewer 2 Report

Dear authors

I would like to congratulate you on this excellent manuscript. The paper is well written. However, the discussion is poor. It needs more studies to compare with other countries and also to check the references of the world food program regarding school lunch program implementation. Also it would be great to talk about the nutrient content and about the importance of school lunch program as tool to prevent from micronutrients deficiencies.   

Author Response

Reviewer #2

1. I would like to congratulate you on this excellent manuscript.

Response: Thank you for your comment and the effort involved in review of the manuscript.

2. The paper is well written. However, the discussion is poor. It needs more studies to compare with other countries

Response: According to the reviewer’s suggestion, we have incorporated more information comparing our results and those of other studies in the discussion section. See page 14  paragraph 4,5  and  page  15  paragraph 1,2

3. Also to check the references of the world food program regarding school lunch program implementation.

Response: Thank you for the comment. In the new version we have made reference to the World Food Program in the discussion.  See page 15 paragraph 2 line 308-3012

4. Also it would be great to talk about the nutrient content and about the importance of school lunch program as tool to prevent from micronutrients deficiencies.   

Response: Taking into account this comment, we have included in the discussion section information on the importance of school food programs to avoid micronutrient deficiency. See page 15 paragraph 1 line 290-292

Reviewer 3 Report

The area of work is significant. The manuscript was well written. A few comments/ suggestions to the authors

 1.     Page 3-18, Results: indicate the geography and school coverages of the 13 guides

2.     Results: Elaborate more for figure 1 as only 1-2 sentences in results and no discussion.

3.     Conclusion: insufficient syntheses; Suggest making bullet points of recommendations. Synthesize the comparisons among indicators and the guidelines – let us know “so what” of the lower/ higher values and the numbers and make recommendations at the ends on for examples, where the guidelines can be further improved or the suitability of WHO/FAO criteria which can be followed or not. The paper’s title is sustainability and nutrition recommendations…..    

Author Response

Reviewer #3

The area of work is significant. The manuscript was well written. A few comments/ suggestions to the authors.

Response: Thank you for the comments and the effort in reviewing the manuscript.

1. Page 3-18, Results: indicate the geography and school coverages of the 13 guides

Response: Thank you for the observation. In the results section, we have named the geographic regions of the guides studied in Spain. See page 4 paragraph 2. Furthermore, in the introduction we have incorporated information on the percentage of the pre-school and primary school population that uses cafeteria services.  See page 2 paragraph 3 line 56-58.

2. Results: Elaborate more for figure 1 as only 1-2 sentences in results and no discussion.

 Response: In response to this suggestion, we have included a brief description of the results in Figure 1. See page 9 paragraph 5 line 176-178.

3. Conclusion: insufficient syntheses; Suggest making bullet points of recommendations. Synthesize the comparisons among indicators and the guidelines – let us know “so what” of the lower/higher values and the numbers and make recommendations at the ends on for examples, where the guidelines can be further improved or the suitability of WHO/FAO criteria which can be followed or not. The paper’s title is sustainability and nutrition recommendations…..  

Response: Thank you for the comment. In response, we have incorporated recommendations in the conclusion in the form of bullet points.  See page 16 paragraph 1 line 330-336
